# Nutrition during Pregnancy and Lactation: Epigenetic Effects on Infants’ Immune System in Food Allergy

**DOI:** 10.3390/nu14091766

**Published:** 2022-04-23

**Authors:** Margherita Di Costanzo, Nicoletta De Paulis, Maria Elena Capra, Giacomo Biasucci

**Affiliations:** Pediatrics and Neonatology Unit, Department of Maternal and Child Health, Guglielmo da Saliceto Hospital, 29121 Piacenza, Italy; n.depaulis@ausl.pc.it (N.D.P.); m.capra@ausl.pc.it (M.E.C.); g.biasucci@ausl.pc.it (G.B.)

**Keywords:** immune tolerance, gut microbiota, dysbiosis, short chain fatty acids, butyrate, vitamin D, vitamin A, polyunsaturated fatty acids (PUFAs), epigenetic mechanisms, breastfeeding

## Abstract

Food allergies are an increasing health problem worldwide. They are multifactorial diseases, in which the genome alone does not explain the development of the disease, but a genetic predisposition and various environmental factors contribute to their onset. Environmental factors, in particular nutritional factors, in the early stages of life are recognized as key elements in the etiology of food allergies. There is growing evidence advising that nutrition can affect the risk of developing food allergies through epigenetic mechanisms elicited by the nutritional factors themselves or by modulating the gut microbiota and its functional products. Gut microbiota and postbiotics can in turn influence the risk of food allergy development through epigenetic mechanisms. Epigenetic programming accounts not only for the short-term effects on the individual’s health status, but also for those observed in adulthood. The first thousand days of life represent an important window of susceptibility in which environmental factors, including nutritional ones, can influence the risk of developing allergies through epigenetic mechanisms. From this point of view, it represents an interesting window of opportunity and intervention. This review reports the main nutritional factors that in the early stages of life can influence immune oral tolerance through the modulation of epigenetic mechanisms.

## 1. Introduction

### 1.1. Food Allergy

Food allergy is recognized as a growing health concern. Over the last several years, epidemiological studies reported a changing pattern of food allergies with not only an increased prevalence, but also an increased persistence and severity of clinical manifestations [1]. Even though genetics contribute to the development of food allergies, the observed epidemiological change cannot be explained by classical genetics, because it is too fast to be justified by changes in the human genome alone. In the current view, the changed pattern of food allergies derives from a complex gene–environment interaction [2]. The effects of the environment on the susceptibility and occurrence of food allergies are realized through epigenetic mechanisms [3]. These mechanisms involve heritable alterations that change gene expression, without altering the DNA sequence [4]. Environmental factors, including nutritional factors, play a crucial role during pregnancy and the first thousand days of life in determining an individual’s susceptibility to developing chronic non-communicable diseases in adulthood, such as allergies [5]. Thus, nutritional factors during the gestational period and breastfeeding can influence immunity through epigenetic mechanisms [6]. Patients with food allergies have a higher risk of other allergies later in life, the so-called atopic march, and other conditions due to a dysregulation of the immune system [7]. Currently, food allergy is considered as the first manifestation of an alteration of the immune mechanisms and derives from a failure in the mechanisms of immune tolerance [8]. For all these reasons, it is very interesting to understand how nutrition in early life can influence the mechanisms of immune tolerance. The aim of this review is to provide an overview of the role of nutrition in early life towards food allergy development, through its epigenetic effects on the immune system of the offspring.

### 1.2. Epigenetic Mechanisms

Epigenetic mechanisms involve transmissible alterations that modify gene expression without altering the primary DNA sequence. These stable alterations are called “epigenetic” because they are inheritable in the short term, but do not lead to DNA mutations and are therefore potentially reversible [9]. Epigenetics regulate gene expression dynamically so that the epigenetic state of a gene can change. This capacity to modify in response to environmental stimuli lies in the post-synthetic modification of the DNA itself (DNA methylation) or of intimately associated proteins (histone modifications) [9]. DNA methylation and histone modifications are the major epigenetic mechanisms recognized so far.

#### 1.2.1. DNA Methylation

DNA methylation involves the addition of a methyl group to cytosine or adenine DNA nucleotides. This epigenetic mechanism results in gene silencing and the inhibition of gene transcription [10]. DNA methylation occurs at the various CpG (cytosine-guanine dinucleotide) sites present as “islands” in most genes. There are several million CpG sites in the genome of a cell, which play a crucial role in the promoter regions of genes. When these sites are methylated, gene transcription cannot take place. Once these sites are methylated, gene transcription may not occur. Conversely, when these CpG sites are demethylated, the promoter region of the gene can interact with the various transcription factors that control gene expression [11]. The main enzymes that regulate DNA methylation are DNA methyltransferases (DNMTs). There are several DNMTs, the main one being DNMT1, which is important for maintaining the methylation status of a gene. Most genes are silenced in their normal state. DNMT3a and DNMT3b are the main enzymes for de novo methylation and mediate methylation-independent gene repression [12]. In T-cells, DNA methylation is implicated in the development, activation, and maintenance of T-cell function. For example, demethylation of the forkhead box P3 protein (FoxP3) may promote the development of regulatory T-cells (Tregs) [13].

#### 1.2.2. Histone Modifications

Histones are highly alkaline proteins located in the nuclei of the eukaryotic cells, where they organize DNA into structural units, which are called nucleosomes. DNA is generally wrapped around two copies of the histones H2A, H2B, H3, and H4 [14]. Post-transcriptional modifications of the protein tails of these histones are the main mechanisms of chromatin regulation [15]. They include the following biochemical process: acetylation, methylation, phosphorylation, ubiquitination, and sumoylation. The results of these modifications range from gene activation to gene silencing and may also have some DNA repair functions [14]. In this regard, histone acetylation of lysine residues is one of the best examples studied, with consequent activation of transcription [14]. This activation is mediated by histone acetyltransferase. The process of histone modifications in T-cells contributes to the differentiation of T-cells into various subgroups [16].

#### 1.2.3. MicroRNAs

Beyond the reported “classical” epigenetic mechanisms, microRNAs (miRNAs) are also widely recognized as significant epigenetic controllers of gene expression [17]. They are small endogenous single-stranded non-coding RNA molecules found in the transcriptome of animals, plants, and some viruses. These are polymers encoded by eukaryotic nuclear DNA, about 20–22 nucleotides long, and mainly involve in regulating gene expression at both transcriptional and post-transcriptional levels. The miRNAs are incorporated into the RNA-induced silencing complex (RISC) and induce gene silencing by overlapping with complementary sequences present on target messenger RNA (mRNA) molecules. This binding results in translation repression or degradation of the target molecule. It is considered that about 60% of the genes coding for proteins are regulated by this mechanism, and therefore, their study constitutes an element of considerable interest in the context of many human diseases, including allergic diseases [18,19].

### 1.3. Pregnancy, Lactation and the First Year of Life: A Window of Opportunity

According to the worldwide-known “Barker’s fetal origins hypothesis”, an insult or an intervention in critical developmental periods can have negative or positive epigenetic effects in terms of functional outcome in the later years of life [20]. The so-called first thousand days of life are considered a delicate and fragile period that can be seen as a “window of opportunity” for the possible protective and health-promoting effects, but also as a “window of susceptibility” for the possible epigenetic negative effect and the major risk of disease development [5]. During this time, nutritional and other environmental factors can influence the risk of food allergies and immune system development through epigenetic mechanisms which interact with genetic susceptibility [21]. For these reasons, modifiable risk factors for food allergies, such as environmental and nutritional ones, have become an interesting target for the prevention of food allergies (Figure 1).

## 2. Early Life Nutrition and Epigenetic Effects in Food Allergy

### 2.1. Gut Microbiota

From early life, the complex relation between diet and gut microbiota significantly influences immunity development and functions. Colonization of the gut microbiota already begins during gestational period, in which it is strongly influenced by maternal nutrition. The gut microbiota of the newborns differs in low diversity and a partial dominance of the phyla Actinobacteria and Proteobacteria [22]. During the postnatal period, nutritional factors continue to affect gut microbiota structure and functions. The gut microbiota of breastfeeding infants is dissimilar to that of formula-fed infants [23]. In fact, breastfeeding is a resource of favorable bacteria species, such as *Bifidobacteria*, *Lactobacilli*, and *Enterobacteriaceae*, and associated benefits for infant health [24,25]. Complementary feeding, because of the introduction of fiber and new protein sources in the diet, is another critical period in which nutritional factors strongly influence the gut microbiota composition and functions. When children start eating solid foods, they have a greater diversity of bacteria species, which include *Enterococci*, *Enterobacteria*, *Clostridium*, *Streptococus*, and *Bacteroides* species [22]. In this period, the gut microbiota shifts toward a more mature community that differs from that of adults, with a greater abundance of bacterial taxa producing short chain fatty acids (SCFAs) [26]. Thus, nutritional factors, together with other environmental factors, such as the type of delivery, use of antibiotics, and lifestyle, greatly affect the development of gut microbiota in infancy [27]. During this period, an impairment in gut microbiota composition and function, named gut dysbiosis, can influence immune tolerance and the risk of developing food allergies [28]. There is not a conclusive relationship between mothers’ diet and atopic manifestations in their children. However, available evidence suggests that a maternal Mediterranean diet, which is abundant in fruits, vegetables, fish, and vitamin D-containing foods, may exert beneficial effects on the offspring, but further evidence is needed to support this hypothesis [29,30]. A more evident association between infant diet and prevention against food allergies has been established instead. An infant diet based on high levels of fruits, vegetables, and home-prepared foods has been related with less food allergy by the age of two years [31]. In particular, Azad et al. demonstrated that low gut microbiota richness and an elevated Enterobacteriaceae/Bacteroidaceae ratio in early infancy are linked with succeeding food sensitization, advising that early gut colonization may influence the onset of atopic diseases, including food allergies [32]. Instead, high sugar and fat use have been linked to an increased risk of atopic manifestations correlated with significant differences in gut microbiota [33,34]. However, various bacteria could influence immune tolerance and no specific bacterial taxa could be associated with food allergy development. An intriguing association between early life nutrition and gut microbiota in food allergy could be represented by the immunomodulatory metabolites of the gut microbiota [35].

### 2.2. SCFAs

The mechanisms by which gut microbiota influences the epigenome are not fully elucidated, though recent findings suggest that some of them refer to postbiotics, bacterial metabolites derived from gut microbiota activity [36]. SCFAs are the main metabolites produced by gut microbiota. SCFAs, mainly acetate, butyrate, and propionate, derive from the fermentation in the colon of non-digestible carbohydrates contained in dietary fiber [37]. The types and quantities of SCFAs vary according to the diet and the gut microbiota. Among SCFAs, butyrate has an important role in gut homeostasis and immune tolerance [35]. It is an interesting link between nutrition in early life and gut microbiota in food allergy development, as shown in human observational studies. Roduit and collaborators evaluated SCFA levels in the feces of twelve-month-old children and concluded that children with the highest levels of butyrate in the feces at twelve months of age had less atopic sensitization, as well as other atopic signs later in their life, such as asthma, food allergies or allergic rhinitis [38]. Studying the fecal microbiome of atopic children in early life, Cait et al. observed a positive relationship between the deficiency of butyrate-producing bacteria at three months of age and the onset of allergic manifestations at later stages of life [39]. The authors also found that infants with allergic manifestations at later stages of their lives lacked genes encoding crucial enzymes for butyrate production [39]. SCFAs can modulate gut epithelial barrier and immunity through different mechanisms. They can act on G protein-coupled receptors (GPRs), such as GPR41, GPR43, and GPR109a, which are present not only on the gut epithelial cells, but also on gut immune cells, such as dendritic cells (DCs) and Tregs [36]. Furthermore, butyrate and propionate can modulate several genes implied in different biological processes, such as gut barrier integrity and immune tolerance to bacterial and food antigens, through the inhibition of histone deacetylase (HDAC) and lysine deacetylase (KDAC) [40,41,42].

Butyrate can modulate the epigenetic status of Tregs, immune cells with a key role in immune tolerance, as it is able to increase the number of intestinal Tregs by stimulating the acetylation of histones at the Foxp3 gene, and also by protecting the Foxp3 protein from degradation through enhancing its acetylation [43,44,45]. Nakajima et al. showed that maternal fiber consumption during gestation and breastfeeding influences plasma levels of SCFAs and Foxp3+ Tregs differentiation in offspring; in this regard, they found that SCFAs plasma levels of offspring of mice fed a highly fiber diet, as well as their number of Tregs, were higher than in mice fed no fiber [46].

Recent evidence suggests that SCFAs are not only HDAC inhibitors but are also able to support histone modifications in immune cells by playing as acyl-CoA precursors. In T-cells, SCFAs increase the activity of mTOR complex by enhancing glucose oxidation and by producing an additional pool of acetyl-CoA [40,47]. Acetyl groups from SCFAs can be ligated to the cellular pool of CoA and the storage of acetyl-CoA in tricarboxylic acid cycle results in higher levels of citrate. The enzyme ATP citrate lyase (ACLY) converts citrate into acetyl-CoA that is used for histone acetylation and regulation of cytokine gene expression [48]. The blockade of ACLY activity results in decreased production of IL-10 and IFN-γ in lymphocytes, both cytokines being involved in the Th1 immune response [49]. Butyrate may promote B-cell differentiation and IgA and IgG production through HDAC inhibition which leads to acetylation of specific genes involved in B-cell differentiation and/or IgG and IgA formation [50].

In conclusion, a high abundance of SCFA-producing bacteria and an adequate consumption of non-digestible carbohydrates, such as fiber, could be required for gut homeostasis and immune tolerance development and maintenance.

### 2.3. PUFAs

Fatty acids with more than one double bond in their biochemical structure are classified as polyunsaturated fatty acids (PUFAs). PUFAs can be divided into two main strands, according to the position of the first double bond: omega-3 (ω-3) and omega-6 (ω-6) [51]. The eighteen carbon atoms precursor compounds of the two series are, respectively, alpha-linolenic acid (ALA; 18:3ω-3) and linoleic acid (LA; 18:2ω-6). As ALA and LA cannot be synthesized endogenously, they must be introduced exogenously with the diet; therefore, they are classified as essential fatty acids. Derivatives with more than eighteen carbon atoms are classified as long chain polyunsaturated fatty acids (LCPUFAs). The most important ω-3 LCPUFAs are eicosapentaenoic acid (EPA) and docosahexaenoic acid (DHA), whereas arachidonic acid (ARA) is the most studied within ω-6 LCPUFAs. DHA and ARA are considered semi-essential or conditionally essential fatty acids, as they can be both assumed with diet and derived endogenously from ALA and LA through the action of delta-5-desaturase and elongase 5 enzymes. PUFAs are precursors of various bioactive metabolites, such as prostaglandins, leukotrienes and thromboxanes [52,53]. The main dietary sources of ALA are linseed oil, wheat germ oil, walnuts, and linseeds, whereas LA is present in high concentrations in sesame oil, sunflower oil, and corn seed oils. Meat and meat-derived products and egg yolk are rich in ARA, whereas fish, fish oils, and eggs are rich in DHA [52] (Figure 2).

Among PUFAs category, ω-3 have an anti-inflammatory action and improve allergic manifestations [54], whereas ω-6 PUFAs are generally considered to have pro-inflammatory effects since they favor Th2 immune response and allergy development [55]. In this regard, the Western diet is characterized by a high dietary ratio of ω-6:ω-3 PUFAs, which are correlated to a major risk of immune-mediated disorders, including food allergy [56].

According to European Food Safety Authority (EFSA) recent recommendations, the Reference Intake of LCPUFAs for pregnant and lactating women should be 250 mg/day of EPA+DHA, of which DHA should account for 100–200 mg/day [57] (Table 1).

Sioen et al. recently analyzed LCPUFAs intake in specific population subsets and they highlighted a suboptimal LCPUFAs intake in pregnant and lactating women [59]. Human milk contains PUFAs; on average, it contains 0.9% wt/wt of ALA and 11% wt/wt of LA [60]. LCPUFAs content of human milk is dependent on mother’s dietary intake [61]. The effect of supplementation with LCPUFAs in pregnancy and lactation has been analyzed in various studies. An increased intake of ω-3 LCPUFAs during pregnancy, with a supplementation of a DHA-rich oil (0.1 g EPA + 0.8 g DHA daily) was linked to reduced sensitization to hens’ egg and a reduced risk of developing an IgE-mediated eczema when compared to the offspring of women who did not receive such supplementation [62]. Studies conducted on pregnant and lactating women have highlighted that ω-3 LCPUFAs, usually the combination of EPA and DHA, exert positive immune effects, even if the duration of these effects is not yet clear, due to the short follow up period reported in studies available in the literature so far [52]. Although the mechanisms through which LCPUFAs influence immune system functions and food allergy development in early life are not fully understood, recent scientific literature suggests that epigenetic mechanisms may play a role. Acevedo et al. demonstrated that fish oil intake in pregnant women is associated with epigenetic regulation of T-cell protein kinase C ζ (PKCζ) and other T-cell-related loci, crucial for T-cell maturation towards Th1 cytokine profile. Specifically, in CD4 + T-cells obtained from cord blood of newborns from mothers supplemented with fish oil during pregnancy, they found a higher acetylation of histone H3; this, in turn, corresponds to a more transcriptionally permissive chromatin status observed at the promoter region of the PKCζ encoding PRKCZ gene. Additionally, they found that the acetylation levels of either H3/H4 or H3 alone at the promoters of IL-13 or T-box 21 (TBX21), respectively, were lower in the fish oil group compared with placebo [63].

In another study, the authors found that H3 acetylation levels at the promoters of FOXP3, IL-10 receptor subunit alpha, and IL-7 receptor genes were significantly increased in the placenta specimens of women who often consumed olive-oil for cooking during gestational period. These results suggest that olive-oil composition can have relevant effects on the histone modifications in placenta [64].

In addition to histone modifications, there is evidence that DNA methylation is another epigenetic mechanism elicited by PUFAs. In this regard, Lee et al. showed that maternal supplementation with ω-3 PUFA during pregnancy may modulate DNA methylation levels of genes encoding Th1 and Th2 cytokines (IFN-γ and IL-13, respectively), thus influencing the Th1/Th2 balance in infants [65].

### 2.4. Vitamin D and A

The studies on the function of vitamin D in food allergies increased in recent years. Vitamin D inadequacy has been correlated to an elevated risk of food allergies, such as peanut and egg allergy [66,67]. Vitamin D can act at different levels to influence immune functions and food allergy development. Available studies suggest that vitamin D status in early life has effects on immune tolerance later in life, probably through epigenetic mechanisms [68,69,70]. Junge et al. reported that high vitamin D levels from cord blood at birth linked with lower levels of methylation in thymic stromal lymphopoietin (TSLP) enhancer region, resulting in an elevated TSLP mRNA expression [71]. TSLP is an epithelial cell-derived cytokine related to allergic diseases. It has been reported to promote Th2 immune response, acting on DCs and basophils [72,73]. Critical vitamin D genes can be silenced by DNA methylation because they contain large CpG sites [69]. It has been demonstrated that severe vitamin D deficiency is associated with methylation changes in leukocyte DNA, in DHCR7, CYP2R1, and CYP24A1 genes, deeply involved in vitamin D metabolism [74]. Maternal vitamin D supplementation during gestation and breastfeeding alters DNA methylation in mothers and breast-fed infants [75]. Furthermore, Jiao et al. showed that vitamin D deficiency during pregnancy causes immune alterations in offspring rats and decreases Th1/Th2 cells ratio and IFN-γ gene production, by methylation of the gene changing DNMT activity [76]. Vitamin D deficiency in pregnant women may be a risk factor for food allergy development also by suppressing Tregs [77]. In mice models, it has been shown that the offspring of mothers fed with a low vitamin-D had an elevated risk of food allergy development, correlated with lower levels of Tregs compared to the offspring of mothers which received a more balanced diet. Insufficient vitamin D levels might lower the Tregs through a change in methylation of FOXP3 and determine an increased susceptibility to food allergy [78]. In contrast, a cohort study demonstrated that high vitamin D levels during gestation and at birth elevated the possibility of food allergy [79]. The relationship between vitamin D and food allergy is not conclusive and future studies are needed to better explore the function of vitamin D in immune tolerance and food allergy.

Retinoic acid (RA), a metabolite of vitamin A, has several effects on immune system development and functions, with effects on innate and adaptive immunity [80]. However, its function in food allergy is not yet clarified. Recently, Maeta and collaborators demonstrated that continuous RA intake under allergen exposition improved the symptoms of food allergy in mice and may promote the induction of anti-inflammatory cytokines, such as IL-10 and IFN-γ [81]. Additionally, butyrate action is mediated by the stimulation of the epithelial RA through the inhibition of epithelial HDAC 3 [41]. Given this, Badolati et al. highlighted RA, butyrate and vitamin D as important modulator and antagonistic factors for Th9 cells, an important subset of cells involved in allergic diseases. In this regard, they supported diet as an innovative strategy to target Th9 cells in the management of allergic diseases [82].

### 2.5. Breastfeeding

Human breast milk should be considered as the “natural” food for newborns and infants; national and international organizations promote exclusive breastfeeding from birth to six months of age and continuation of breastfeeding throughout the complementary feeding period [83]. Human breast milk has many functional effects, as it exerts positive epigenetic effects on the infant, with an overall protection towards the development of non-communicable diseases [84]. Breastfeeding has an anti-inflammatory effect on the infant’s immune system [85]. Saarinen et al. and Muraro et al. conducted studies that highlighted the protecting effect of breastfeeding on atopic eczema and food allergy [86,87]. In the Melbourne atopy cohort study, human milk oligosaccharides profiles were linked to allergic disease risk in childhood [88]. The mechanisms underlying breast milk positive epigenetic effects in allergic diseases are partly unknown. It is hypothesized that DNA methylation, histone modifications and miRNAs are modulated by nutritional exposure [5]. Recent studies have shown that the gene encoding for sorting nexin 25 (SNX25) can be involved in this process. The increase of promoter methylation reduces gene expression and consequently down-regulates the protein level of SNX25. SNX25 can down-regulate TGF-β, that is a component of breast milk implied in the pathogenesis of allergic diseases [89].

Human breast milk is very rich in miRNAs, which represent one of the most important epigenetic mechanisms to explain its beneficial effects in the breast-fed infant. miRNAs can affect gastrointestinal and immune system development of newborns. They can act as potential immune protectors for the infant and the mother, by the regulation of T and B cell development, differentiation of DCs and the production of inflammatory cytokines [90]. Simpson et al. demonstrated that, three months postpartum, human breast milk has a group of highly expressed miRNAs, as miR-148a-3p, miR-22-3p, miR-30d-5p, let-7b-5p and miR-200a-3p. These miRNAs are involved in different biological processes and molecular functions.

SCFA butyrate is another bioactive compound of human breast milk that can protect against food allergy development. It can impact on different processes involved in immune tolerance to foods. In an animal study, pre-treatment with SCFA butyrate can reduce allergic responses thanks to an increased expression of tolerogenic cytokines, a reduced Th2 cytokine production, and a regulation of oxidative stress [91]. In addition, this study showed that SCFA butyrate up-regulate mucin, tight junctions, and human beta defensin-3 expression in human enterocytes [91]. The analysis of peripheral blood mononuclear cells (PBMCs) from children with food allergy highlighted that butyrate increases IL-10, IFN-γ, and FOXP3 expression by epigenetic modifications. Moreover, it supported DCs, Tregs, and the precursors of M2 macrophages [91].

In conclusion, new strategies to improve the protective effect of human milk against food allergy development may be based on the modulation of maternal diet to increase bioactive compounds in human milk.

### 2.6. Complementary Feeding

Regarding complementary feeding and food allergy prevention, current international guidelines do not recommend the delayed introduction of solid foods in infancy because it has showed to be ineffective in food allergy prevention. On the other hand, there are no data to support early exposure before four months of age [92,93,94].

As mentioned above, complementary feeding is a key period in which foods influence the gut microbiota [22,26]. Through the modulation of the gut microbiota and its products, complementary feeding can intervene in the risk of allergies eliciting epigenetic mechanisms. Currently, there are no available studies on the epigenetics of complementary nutrition and the risk of food allergies.

## 3. Conclusions

This review focused on the potential effects of early life nutrition in epigenetic processes and its consequences on food allergy development. Nutritional factors, like environmental ones, can affect the development of the immune system and food oral tolerance in the early stages of life, through epigenetics. DNA methylation, histone modifications, and miRNAs are epigenetic mechanisms that can modify the risk of allergic diseases, such as food allergy.

The first thousand days of life represent a “window of opportunity” to modify the risk of developing non-communicable diseases, such as allergic diseases, and nutritional factors represent an interesting target of intervention. This aspect is not easy to investigate, both from a qualitative and quantitative point of view, considering that there are many environmental factors that influence epigenetic programming in this period of life. Future studies in this area are desirable to better clarify epigenetic mechanisms elicited by nutritional factors in the risk of food allergy and to develop targeted intervention strategies for prevention and treatment.

## Figures and Tables

**Figure 1 nutrients-14-01766-f001:**
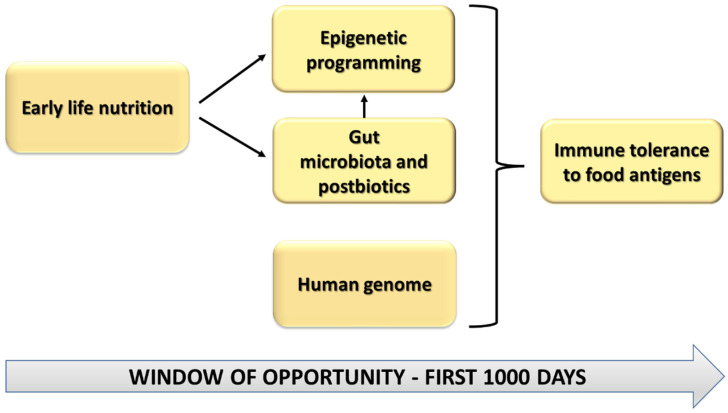
Interrelation among early life nutrition, human genome, gut microbiota and postbiotics, and epigenetics during the first 1000 days of life.

**Figure 2 nutrients-14-01766-f002:**
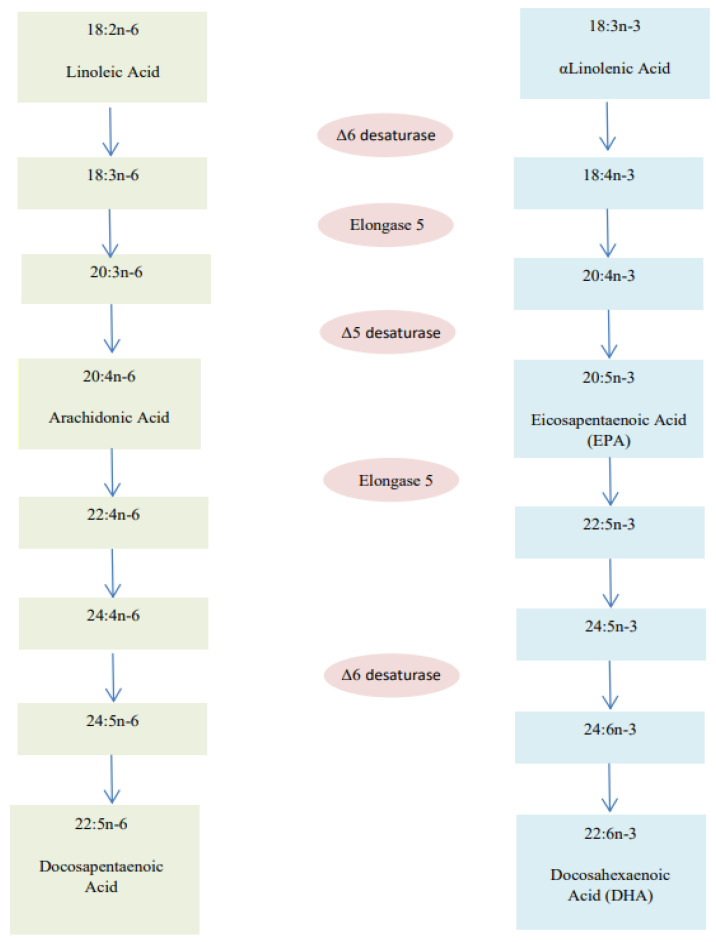
Pathway of long-chain polyunsaturated fatty acid (LCPUFAs) biosynthesis from essential fatty acid precursors.

**Table 1 nutrients-14-01766-t001:** Daily reference amounts of nutrients involved in food allergy modulation in critical development periods, according to LARN 2014 [58].

	Pregnancy	Lactation	Infants 6–12 Months
Vitamin D	15 µg	15 µg	10 µg
Vitamin A	1000 µg	700 µg	450 µg
LCPUFAs	EPA + DHA 250 mg + DHA 100–200 mg	EPA + DHA 250 mg + DHA 100–200 mg	EPA + DHA 250 mg + DHA 100 mg
Fibers	12.6–16.7 g/1000 kcal	12.6–16.7 g/1000 kcal	8.4 g/1000 kcal

Population Reference Intake (PRI) and/or Adequate Intake (AI) and/or Reference Intake (RI) daily amounts of vitamin D, vitamin A, LCPUFAs, and fibers. Vitamin D is expressed as cholecalciferol (1 µg cholecalciferol = 40 UI vitamin D). Vitamin A is expressed as µg of retinol activity equivalents (RAE).

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
