# Peer review of "Nutrition during Pregnancy and Lactation: Epigenetic Effects on Infants’ Immune System in Food Allergy"

_nutrients, 2022, doi:10.3390/nu14091766_

Round 1

Reviewer 1 Report

The paper entitled “Nutrition during pregnancy and lactation: epigenetics effects on immune system in food allergy” addresses an interesting issue that needed to be revised.

It is a well written review about the effect of “nutrients” and their influence during pregnancy and lactation on the offspring immune system and its impact on the future progression to food allergy. It summarizes how epigenetic factors during this specific window of development can be involved in the health of the future adult and how this is dependent of the maternal dietary intake (quality and frequency).

Is concise, to the point. Overall is a clear review of an interesting topic, although there is still a lot of unknowns that needs further research.  I have no further revision comments.

Author Response

Point 1: The paper entitled “Nutrition during pregnancy and lactation: epigenetics effects on immune system in food allergy” addresses an interesting issue that needed to be revised.

It is a well written review about the effect of “nutrients” and their influence during pregnancy and lactation on the offspring immune system and its impact on the future progression to food allergy. It summarizes how epigenetic factors during this specific window of development can be involved in the health of the future adult and how this is dependent of the maternal dietary intake (quality and frequency).

Is concise, to the point. Overall is a clear review of an interesting topic, although there is still a lot of unknowns that needs further research.  I have no further revision comments.

Response 1: Thanks for your positive comments.

Reviewer 2 Report

The authors have done well aligning/linking antenatal nutrition with childhood allergy and intolerance.  The paper is well-written. I only have some few suggested edits.

  1. Title: Edit the title to indicate what is being affected; the mother or infant? I guess it is epigenetic effects on infants?
  2. Overall, a clear case for a link between maternal nutrition during pregnancy and lactation has only mildly been made.
  3. Some information on inappropriate weaning and food allergy was in this paper, not evidently made
  4. Any information on the contributions of complementary foods to allergies?
  5. Line 217: If using ALA is for Alpha-lenolenic acid, not just lenolenic acid.
  6. Figure 2 should be referenced
  7. Figure 2 does not add extra information so can be deleted to save space.
  8. Table 1 footnote: Vitamin A is in RAE, not RE. Please correct this.
  9. Table 1: Why are the fiber values underlined? Add why as footnote if they need to be underlined.

Author Response

The authors have done well aligning/linking antenatal nutrition with childhood allergy and intolerance.  The paper is well-written. I only have some few suggested edits.

Point 1: Title: Edit the title to indicate what is being affected; the mother or infant? I guess it is epigenetic effects on infants?

Response 1: As suggested, we edited the title as follows: “Nutrition during pregnancy and lactation: epigenetic effects on infants’ immune system in food allergy”.

Point 2: Overall, a clear case for a link between maternal nutrition during pregnancy and lactation has only mildly been made.

Response 2: The aim of this review is to provide an overview of the role of nutrition in early life towards food allergy development, through its epigenetic effects on infants’ immune system. We reported the effects of the nutrients during gestational period and breastfeeding divided into the paragraphs.

Point 3: Some information on inappropriate weaning and food allergy was in this paper, not evidently made

Response 3: We added a paragraph about complementary feeding.

Point 4: Any information on the contributions of complementary foods to allergies?

Response 4: We added a paragraph about complementary feeding.

Point 5: Line 217: If using ALA is for Alpha-lenolenic acid, not just lenolenic acid.

Response 5: ALA stands for alpha linolenic acid. We have specified this aspect in the text and in the list of the abbreviations.

Point 6: Figure 2 should be referenced.

Response 6: Figure 2 is adapted from reference number 52. This reference has now been specified in the Figure legend.

Point 7: Figure 2 does not add extra information so can be deleted to save space.

Response 7: It is true, but we thought that a remind of LCPUFAs biochemical pathway could be useful for readers. We refer the decision to remove it or not to the publisher.

Point 8: Table 1 footnote: Vitamin A is in RAE, not RE. Please correct this.

Response 8: Ok, vitamin A is expressed in RAE, we have corrected it in the footnote.

Point 9: Table 1: Why are the fiber values underlined? Add why as footnote if they need to be underlined.

Response 9: The fiber values should not be emphasized. We have standardized the formatting of the table text.

Reviewer 3 Report

The manuscript describes the main nutritional factors that in the early stages of life can influence immune oral tolerance through the modulation of epigenetic mechanisms. The topic is not entirely new and there are some recent reviews addressing the same topic. Still, the manuscript has its merit, it is very well written and structured, and I suggest authors try to give a novel addition to the existing reviews. For example, this manuscript could highlight the major gaps and difficulties in getting evidences of the modulation of epigenetic mechanisms, giving a critical holistic overview on their impacts on food allergy development. Please consider adding.

Author Response

Point 1: The manuscript describes the main nutritional factors that in the early stages of life can influence immune oral tolerance through the modulation of epigenetic mechanisms. The topic is not entirely new and there are some recent reviews addressing the same topic. Still, the manuscript has its merit, it is very well written and structured, and I suggest authors try to give a novel addition to the existing reviews. For example, this manuscript could highlight the major gaps and difficulties in getting evidences of the modulation of epigenetic mechanisms, giving a critical holistic overview on their impacts on food allergy development. Please consider adding.

Response 1: We specified in the conlusions that: This aspect is not easy to investigate, both from a qualitative and quantitative point of view, considering that there are many environmental factors that influence epigenetic programming in this period of life. Future studies in this area are desirable to better clarify the role of epigenetic mechanisms elicited by nutritional factors in the risk of food allergy and to develop targeted intervention strategies for prevention and treatment.